# Vital Block and Vital Sign Server for ECG and Vital Sign Monitoring in a Portable u-Vital System

**DOI:** 10.3390/s20041089

**Published:** 2020-02-17

**Authors:** Tae Wuk Bae, Kee Koo Kwon, Kyu Hyung Kim

**Affiliations:** Daegu-Gyeongbuk Research Center, Electronics and Telecommunications Research Institute, Daegu 42994, Korea; kwonkk@etri.re.kr (K.K.K.); jaykim@etri.re.kr (K.H.K.)

**Keywords:** vital sign, IoMT, patient monitoring device, healthcare, wireless

## Abstract

An important function in the future healthcare system involves measuring a patient’s vital signs, transmitting the measured vital signs to a smart device or a management server, analyzing it in real-time, and informing the patient or medical staff. Internet of Medical Things (IoMT) incorporates information technology (IT) into patient monitoring device (PMD) and is developing traditional measurement devices into healthcare information systems. In the study, a portable ubiquitous-Vital (u-Vital) system is developed and consists of a Vital Block (VB), a small PMD, and Vital Sign Server (VSS), which stores and manages measured vital signs. Specifically, VBs collect a patient’s electrocardiogram (ECG), blood oxygen saturation (SpO2), non-invasive blood pressure (NiBP), body temperature (BT) in real-time, and the collected vital signs are transmitted to a VSS via wireless protocols such as WiFi and Bluetooth. Additionally, an efficient R-point detection algorithm was also proposed for real-time processing and long-term ECG analysis. Experiments demonstrated the effectiveness of measurement, transmission, and analysis of vital signs in the proposed portable u-Vital system.

## 1. Introduction

IoMT represents a newer technology that signifies the applicability of Internet of Things (IoT) into the medical or healthcare industry. Specifically, it implies connecting smart medical devices with in-built sensors and actuators that can record medical data in real-time and sharing medical data through the internet over another device [1]. It exhibits high applicability to healthcare systems. The following tasks require high usability of IoMT; newer product development, real-time data generation, treatment adherence monitoring, a smarter medical decision, improved healthcare infrastructure, and customized product development and care [2]. Originally, IoT represents a network of physical devices that is embedded with electronics, software, sensors, and network connectivity, which enables the objects to transfer data [3]. A report predicted that 40% of IoT-related technology is health-related and will constitute a $117 billion market by 2020 [4]. The IoMT market is expected to grow at a compound annual growth rate of 30.8%, from $41.2 billion in 2017 to $158.1 billion by 2022 [5]. The convergence of medicine and information technologies is expected to transform the current state of healthcare, thereby curbing costs, reducing inefficiencies, and saving lives. The growth is due to the rapid digitization of healthcare systems to aid efficient patient care, rise in demand for mobile healthcare technologies, and increased demand from aging population and people suffering from chronic diseases [6].

IoMT based wearable devices can monitor long-term bio-signals and upload biometric data to the cloud via a wireless communication system [7]. They can collect the sensor data to monitor basic vital parameters in wireless body sensors or patient’s smartphone using high-speed Bluetooth [8]. The collected vital data is temporarily buffered in the mobile device and can also be stored in a database. The vital data can be analyzed in the mobile device itself to verify the warning level. If the analyzed result exceeds a threshold, then a warning alarm is sent to medical staff. The warning alarm and temporarily saved vital data are transmitted to a central database situated in the hospital. If a doctor needs to see the patient’s electrocardiogram (ECG) signal in his or her own phone, then the system can provide a provision to view the ECG report on the phone.

ECG corresponds to one of the most representative biometric parameters. In remote ECG monitoring, QRS detection is the most important step to understand ECG heartbeat including arrhythmia. The QRS detection method capable of real-time processing with excellent performance corresponds to the key technology in the ECG analysis algorithm. However, measured ECG signals typically exhibit very different characteristics based on a patient due to noise or motion artifacts, and this leads to difficulty in detecting the QRS complex. Many QRS or R-point detection methods were based on derivative filter [9], Hilbert transform [10], adaptive threshold [11], mathematical morphology [12], multiple filters [13], wavelet transform [14], and temporal energy [15]. Although the aforementioned methods exhibit excellent performance in the QRS detection, they involve high computational complexity in real-time or long-time ECG analysis. In the study, we introduce a novel efficient QRS detection method that can process the heart rate variability (HRV) analysis in real-time.

The proposed portable u-Vital system introduced in the study mainly consists of a Vital Block (VB) (which is a small patient monitoring device (PMD) that is capable of obtaining patient vital signs in real-time) and Vital Sign Server (VSS) (which transmits, analyzes, and manages the data wirelessly). The VSS creates a database including patient vital signs, performs long-term analysis of vital signs, and ensures that the patient records are secure through a database lock. A VB continuously monitors patient vital signs in the ward and recovery room or measures changes in the patient’s condition during medical procedures in the emergency room and operating room. Additionally, it sends measured vital sign data to smart devices or central monitoring system. The vital sign data transmitted to a VSS can be analyzed to aid clinicians in assessing the patient’s condition and performing appropriate treatment. In case a patient’s condition is continuously recorded, such as in a gastrointestinal endoscopy center or an artificial kidney room, it can aid in patient care by eliminating the labor-intensive work of medical staff.

## 2. Vital Sign Sensing Device Trend and Data Transmission

An advance in healthcare incudes the ability to more precisely monitor patients in a facility’s care and discharged patients. Remote patient monitoring allows medical professionals to monitor vital signs and assess bodily reactions to treatments given to each individual without the need to be in the same physical location as the patient. Vital sign sensing devices can measure various biometric parameters, such as ECG, SpO2, NiBP, BT, muscle electromyography signals, glucose levels, galvanic skin response, lung capacity, and body scale parameters [16,17]. They can also include various wireless personal area network (WPAN) interfaces, such as ZigBee/IEEE802.15.4, Bluetooth BLE, and 400MHz RF, based on their function [18,19,20]. Recent vital sensing devices use mobile OS (such as Android/iOS) connectivity to send all the data to the smartphone [21,22]. Additionally, they also allow access to the history of the gathered data by connecting to the cloud through a web browser or using native mobile apps [23,24,25].

Recent trends in the technology of PMDs are as follows. First, the demand for new technologies is increasing, and strategic alliances and mergers are underway to develop new markets. For example, Philips Healthcare (one of the world’s leading providers of PMDs) acquired Respironics (an end-inspiratory CO_2_ module company) to complement the scarce technology sector and also acquired Goldway (a low-cost PMD company) to expand low-cost patient monitoring system (PMS) market [26]. Second, the share of medical institutions in global companies is increasing. Global top companies are blocking the entry of latecomers via selling integrated equipment that includes PMDs and also medical devices used in medical institutions through the hospital information system [27,28]. Second-ranked companies that develop, produce, and sell only PMDs fail to narrow the gap with first-rate companies and prepare self-rescue measures such as original development manufacturing and original equipment manufacturing for PMDs of first-rate companies [29]. Third, IT technology is combined with traditional patient measurement devices to develop a healthcare information system. [30]. Given the introduction of IT technology and the various needs associated with patient information, standardization of patient data is under progress [31,32]. Furthermore, the collected patient data is developed such that it can be checked anytime and anywhere through the entire network of medical institutions. Fourth, the development of noninvasive measurement devices is required. There is a need for non-invasive and accurate measurements in bi-spectral index (BIS) for anesthesia level, blood pressure, and blood sugar [33].

The classification based on the performance of the PMS is as follows. First, low-performance PMDs [34] are devices that measure and record the most basic biological signals, such as heart rate (HR), NiBP, and BT, which are typically used in hospitals when low-level monitoring is necessary or it is not urgent (typically used in an ambulance or small hospital). Second, middle-performance PMDs [35] generally add measurements of SpO2 and respiratory rate (RR) to low-performance PMDs (used in hospitals, emergency rooms, and recovery rooms). Third, the high-performance PMD [36] is a device that adds BIS monitoring and ECG output to the middle-performance PMS (used in intensive care units and operating rooms. Fourth, the central PMS refers to a system that displays and controls patient data from individual PMDs based on LAN.

The research and development trends of PMDs are as follows. First, icuPATCH (Heart Sounds, Inc., Chicago, IL USA) is developed for cardiovascular monitoring for patients in an intensive care unit and is designed to measure pulmonary arterial pressure noninvasively [37]. Additionally, computer-based stethoscopic techniques are also applied to measure cardiovascular health. Second, the monitoring instrument for cardiology (MONICARD) is developed to measure ECG, SpO2, NiBP, and body weight automatically via voice commands. Additionally, it can measure the risk indexes of pulse transit time and congestive heart failure [38]. It includes an alarm function for patients taking the drug, and the measured bio-signal is encrypted and transmitted to the MONICARD data-center. Third, Nemo Patch (Nemo Devices AG) is designed to monitor blood flow and oxygen in the brain by applying infrared spectroscopy technology. Fourth, vital sign detection device (Pneumo Sonics, Inc., Connecticut, CT, USA) is a remote, portable, and low-power device to detect heartbeat and respiration without skin contact and can act as an electric stethoscope for cardiac and respiratory vital signs.

## 3. Proposed Portable u-Vital System

### 3.1. Configuration of VB and VSS

The developed portable u-vital system uses the VBs (small PMDs) to continuously monitor the patient vital signs in a hospital ward or recovery room and sends the measured vital signals to a smart device or central monitoring system. At least 48 VBs can be connected to one VSS and one VB is allocated per patient. Furthermore, it constitutes a system that helps medical staff to diagnose the current condition of a patient and to perform the appropriate treatment by storing real-time bio-signals of the patient on a VSS. The key devices of the portable u-vital system are composed of a VB for measuring and observing a patient’s vital signals in real-time and a VSS for storing and managing the vital signals measured from the VB. VBs collect patient information, such as ECG, SpO2, NiBP, and BT, in real-time. The collected biometric information is transmitted wirelessly (via WiFi) to a VSS and stored.

The connection configuration of the VB and VSS is shown in Figure 1. A VSS is a core device of the system that transmits and stores biometric data measured in real-time from vital sensors (connected to the VB) and relays the data to each connected GUI client (on Android or Window OS). The main functions include GUI client network connection management with the VB and storage (or management) and analysis (or alarm) of measured data. The real-time vital sign GUI client corresponds to a module that relays and visualizes measured data in real-time from a VSS and exposes the data to users. The users can use the client program to monitor data collected in real-time and generate a report on the analyzed data.

The VSS to measure real-time bio-signals and analyze an ECG signal can connect up to 32 VBs and process the biometric information collected from the VBs without loss. The data measured from the VBs are transmitted to the VSS over WiFi. Additionally, user authentication and password management functions are provided when accessing the VSS via VB or smart devices. The bio-signal information received from the VB is output to a monitor in real-time by the UI client software and stored in MySQL database. Old biometric information is automatically removed, and removal rules can be set. The VSS runs for 24 h in a day, and the OS corresponds to Window Server Standard 2012 R2. The VB interface server software and UI client software are created as separate files and mounted on separate computers.

### 3.2. Flow Chart of VB and VSS

The developed application consists of a central monitoring system (CMS) program in the VSS and viewer program in the Android smart device. The CMS program uses dual-monitors. The main-monitor is used for the central monitoring of transmitted bio-signals and the sub-monitor is used for data processing (DP). An overview of the user flow diagram for the dual-screen configuration of the CMS program is shown in Figure 2. Among the subscribed users, administrators and medical staff can access the CMS and DP screens. On the CMS screen, a patient window with measured bio-signals of each patient is observed. Additionally, trends and analyses of collected bio-signals (especially for ECGs) are seen on the DP screen.

Figure 3 shows the user flow diagram of the viewer program in an Android smart device. If the logged-in user is a patient carer, only two menus (Real-time view and Trend view) are available for providing basic bio-signal information measured for the patient. Conversely, if the logged-in user corresponds to a medical staff, six menus (Patient selection, Real-time view, Alarm view, Replay, Trend view, Long-term ECG analysis) can be selected.

### 3.3. GUI of VB and VSS

As mentioned above, the server program is divided into the CMS program and DP program. First, as shown in Figure 4 the CMS program includes a patient window in the center of the screen where information for up to 32 patients can be viewed in real time. There are multiple patient windows on a page with multiple split screens. The user can double-click on the patient window of interest to view more detailed information on the DP monitor. The user can also check the connection status with an icon in the toolbar at the top of the screen. A patient window supports two modes, patient monitor (PM) mode and patient data terminal (PDT) mode, and one of the modes is displayed on the screen based on the mode set in the corresponding VB. As shown in Figure 4a, the PM mode receives bio-signals, such as ECG, measured in the VB from the server and displays them in real-time. The PDT mode is used to receive information from the VB via the VSS or to provide information to the VB. The PDT mode displays the patient name, VB equipment name, room name, and bed number registered in the current patient window at the top left of the screen, as shown in Figure 4b, and transmits the information that the user wants from the VB to the VSS.

The DP program includes several functions (Patient Management, Connect Patient, System Setup, Vital Sign Trend, ECG Analysis, and Data Retrieve), as shown in Figure 5. First, the Patient Management menu can search for registered patients or select a specific patient among the searched patients. The Connect Patient menu consists of Patient, Alarm, and Wave sub-menus. Basic information on the patient is displayed in the Patient sub-menu. In the Alarm sub-menu, when the patient window of the connected patient is selected on the CMS monitor screen, the alarm information (upper and lower limits of HR, NiBP, BT, SpO2, and RR) of the corresponding VB is initially set in each item of the screen. The System Setup menu is configured by receiving the IDs of all VBs from the VSS and consists of VB, Alarm, User, Hospital, and Language sub-menus. The Alarm sub-menu among the sub-menus is initially configured by receiving existing alarm information from the VSS. The User sub-menu can search for registered users or select a specific user from the searched list. The Hospital sub-menu receives and configures all hospital information from the VSS. The Vital Sign Trend menu consists of the Alarm and Trend sub-menus. In the Alarm sub-menu, if a user enters patient information and a search term, then alarm lists corresponding to the conditions appear in the list below. When a specific alarm among the searched alarm lists is selected, the vital signals at the time before and after the corresponding alarm is generated are displayed as a chart on the right side of the screen. In the Trend sub-menu, when patient information and search period are entered, bio-signal information corresponding to the conditions is displayed at the bottom of the screen. The ECG Analysis menu consists of Report, Heart Rates, RR Interval, ST-Segment, QT-Interval, and Significant ECG sub-menu. In the Report sub-menu, if the patient information and the search term are entered, then the use-list of the VB corresponding to the condition is searched. When a specific item among the searched lists is selected, an ECG analysis result for the data is displayed. The Heart Rates sub-menu calculates minimum, maximum, and average HR for the retrieved ECG data, and the retrieved ECG data is displayed at the bottom. The RR Interval sub-menu calculates the RR interval of the selected ECG data to estimate tachycardia, bradycardia, histogram, and FFT spectrum. The Significant ECG Event sub-menu displays the time zone in which arrhythmia occurred for the selected ECG data. When a specific time zone is selected, ECG data including the corresponding arrhythmia is displayed.

### 3.4. Usage Scenario between VB and VSS

The VB and VSS communicate via the transmission control protocol (TCP) and all packets used in the system are transmitted and received based on Little-endian. The data of the system (i.e., the measured bio-signal data) is stored in the MySQL database [39] and is linked with the hospital electronic medical record (EMR) system. The database field layout of the CMS is decoded via the proxy server located at the hospital firewall and transferred to the Oracle DB of the hospital EMR server, as shown in Figure 6.

Figure 7 shows the flow chart based on the usage scenario between the VB and VSS. The usage scenario represents a network flow diagram from the VB login request to the measured data transfer of the VB. The general usage scenarios are as follows. First, the VSS opens listen socket on port 9999 and listens (waits) for the connection request. Additionally, the VB requests a TCP-based connection to the listening port of the VSS. The VB requests a login to the VSS after the TCP connection is successfully accepted. After the login request of the VB is successfully processed in the VSS, the VSS responds to the VB that the login request was successful. Following a successful login request, the VB sends the patient ID entered by a medical staff to the VSS. When the patient ID is normally received and processed in the VSS, the VSS accepts a successful login request to the VB. After the login request and the patient ID transmission to be measured are successful, the VB transmits the measured bio-data to the VSS based on a predetermined cycle (The bio-data transmission/reception time between the VP and VSS = 0.5 s). The VSS responds with a successful data transfer to the VB after the measured data of the VB is normally sent to the VSS. As the vital sign measurement devices continue the measurement activity, the data transfer from the VB to the VSS and success response from VSS to the VB are repeatedly performed. When the VB is logged in and the patient to be measured is set, the VSS relays the measured data to the logged-in PC and tablet PC (or smart device) client. Among the specific VBs that are logged in, the VBs that are disconnected (socket disconnect) from the network are logged out. It is noted that when a patient’s information changes (such as room number or bed position), the VSS forwards the modified patient information to the VB associated with the patient ID.

### 3.5. Efficient R-Point Detector for Real-Time ECG Processing

In typical ECG signals, the QRS complex is narrow and high while the P or T waves are gentle. Therefore, the detector to detect only the R-point in the P and T wave should consider the characteristics of the R wave. The proposed R point detector is designed to exhibit a similar detection performance for R waves with normal, flat, and spike peaks, as shown in Figure 8.

The R wave is enhanced via the triangulation of ECG sample points. The triangulation T(t) for a sample point t using (t−b, I(t−b)), (t, I(t)), and (t+b, I(t+b)) is calculated as follows:(1)$$T(t)=\frac{1}{2}\left|\left(\left(t-b)\times I(t)+t\times I(t+b)+(t+b)\times I(t-b\right)\right)-\left(\left(t-b)\times I(t+b)+(t+b)\times I(t)+t\times I(t-b\right)\right)\right|,$$
where *b* denotes the half width of the triangulation, and *b* = 1 is used in the study.

Figure 9 shows the block diagram for the proposed R-point detection method. The proposed method applies a max-filter to (normalized) the triangulation values for an original ECG signal to enhance the R-wave interval. The max filter defines a candidate interval of the R-point detection for the enhanced R wave by the triangulation processing. The size of the max filter used for MIT-BIH arrhythmia database (360 samples/sec) [40] is 1 × 25 sample interval. Thereafter, the adaptive threshold is applied to the sliding window to detect a section in which the R-point exists. The size of the sliding window and step size were set to 4000 and 500 samples respectively based on MIT-BIH arrhythmia database. The adaptive threshold is defined as the average of the top 40% of the max-filtered values for triangulation values sorted in descending order relative to the sliding window. Subsequently, if the maximum value in each R-point presence interval exceeds the average of the sample points of the sliding window, then it is detected as an R-point. Otherwise, the minimum value is detected as an R-point for negative R-point such as premature ventricular contraction (PVC).

Figure 10 shows the triangulation, max-filter, adaptive threshold, and R-point detection result for various records of MIT-BIH arrhythmia database. In the case of normal sinus rhythm of Record 234, only the QRS complexes are enhanced, and the R-points are normally detected. The Left bundle branch block (LBBB) rhythm of Record 111 includes spiked R peaks and peaked T waves, while the Right bundle branch block (RBBB) rhythm of Record 118 includes spiked R peaks and inverted T waves. In the triangulation of these two rhythms, the T-waves are suppressed while the QRS complexes are enhanced. The Record 109 also includes F beats in multiple LBBBs in the sliding window. Specifically, F beat denotes the fusion of ventricular and normal beat. Although the triangulation value of the F beat exceeds that of the LBBB beats, the threshold is adaptively adjusted such that all the R-points in the LBBB beats can be accurately detected.

The false negative (FN) and false positive (FP) are applied to estimate the performance indexes of the proposed method. The performance indexes include sensitivity (Se [%] = TP/(TP + FN)), positive prediction (+P [%] = TP/(TP + FP)), and detection error rate (DER [%] = (FP + FN)/(TP + FN)). The R-point detection results of the proposed method via the MIT-BIH arrhythmia database is shown in Table 1. Some signals, such as Record 108, 203, and 210, has a high false alarm rate because they have many arrhythmic rhythms such as atrial flutter, atrial fibrillation, ventricular trigemini, and ventricular tachycardia. On the other hand, normal signals without arrhythmia rhythms such as Record 100, 102, and 103 have very high sensitivity and positive prediction. The values of the proposed method correspond to Se = 99.81%, +P = 99.83%, and DER = 0.36% for the MIT-BIH arrhythmia database. Table 2 lists the performance index comparison of the proposed method and existing methods for total beats (TBs) of all the Records of the MIT-BIH arrhythmia database. The red numbers in the table denote the values of existing methods that are superior to the proposed methods. The proposed method is ranked 3rd among the 8 methods and exhibits a high Se and +P but a low DER in manner similar to existing methods. 

## 4. Results

### 4.1. Developed VB and VSS

The developed vital sign measurement devices (Vital Patch (VP) and VICON), VB, VSS, Android App (downloaded in Google play), and PC Software are shown in Figure 11. Dell Precision T3620 (Intel Xeon E3-1224, 8 GB, Window Server 2012) is used for the VSS. The VSS can connect up to 32 VBs and process vital signals obtained from up to 8 VBs without loss. It also includes the aforementioned CMS and long-term ECG analysis function. The database management system (DBMS) was developed to store, search, and delete vital signals received from the VBs. To obtain the vital signals of a patient, the VB consisted of a 1-channel (Lead II) ECG module to obtain ECG signals, SpO2 module, NiBP module, BT module, and LCD module to display the measured vital signal information. SpO2 module and NiBP module among these modules are connected to VICON. Furthermore, WiFi and Bluetooth modules for data transmission to the VSS were installed in the VB for management and storage of obtained vital signals, and the firmware for controlling each module was also developed. The VP measured the patient’s ECG via the Lead II channel, and the measured ECG signal can be viewed in the smart device or VBs via Bluetooth. The VICON was connected to NiBP, SpO2, and BT measurement modules. The proposed portable u-Vital system consists of VBs and VSS and is connected to medical measurement devices and EMR via Bluetooth to a WiFi bridge, as shown in Figure 12. The system response time is less than 6 s. The data transmission/reception time between the VP and VSS as well as VICON and VSS is less than 1 s.

Many patient monitoring systems currently use different data formats, making medical data less compatible. Therefore, standardization of medical data is urgent as mentioned Section 2. The developed system also uses the data format we defined. Figure 13 shows an example of the data format of the proposed u-Vital system. One bit is assigned to Event, PVC, Bradycardia, Tachycardia, and R point flags, and 2 bits are assigned to Lead-off. In addition, 7 bits for battery information, 11 bits for heartrate and RR interval, and 24 bits for ECG data are allocated.

### 4.2. ECG Experiment for Proposed R-Point Detection Method

Actual ECG data was measured from the VP with a 250 Hz sampling rate and analyzed throughout HRV by the proposed R-point detection method. Figure 14 shows the R-point detection result for a non-patient jogger, a ventricular bigeminy patient, and an atrial fibrillation (AF) patient under antiarrhythmic medication. Although the ECG signal of the jogging contained noise due to motion, it was observed that the proposed method accurately detects R-points. In its triangulation, the P and T waves were suppressed while the QRS complexes were enhanced. As shown in the figure, the ventricular bigeminy ECG signal represents an ECG signal in which the short and long RR intervals are periodically repeated. The bigeminy’s ectopic beats mainly corresponded to PVC. Furthermore, the difficulty in detecting the R-point in the bigeminy ECG signal was due to two types of PVC with different depths. It was observed that the threshold for the R-point detection was lowered by the proposed adaptive threshold method when compared to that of the jogging ECG signal to also detect the PVC signal with a shallow depth. The proposed method detected the R-points for two different types of PVCs. The AF patient consumed antiarrhythmic medicine, and thus the RR intervals remained normal while the signal still contained AF-specific oscillation and large T waves. In the triangulation, the large T waves were effectively suppressed while the QRS complexes were noticeable. Thus, it was confirmed that the R points were accurately detected.

Figure 15 shows the HRV analysis of the jogging signal for 2 min 30 s, ventricular bigeminy signal for 30 min, and AF signal for 3 min. The analyzed HRV included beat per minute (BPM) and trend, histogram, and power spectral density (PSD), and poincare plot for RR interval. 

The bigeminy signal with a long signal length produced multiple BPMs while the jogging and AF signals with very short signal lengths generated only two BPMs. First, the jogging signal exhibited a high BPM due to the increased cardiac output albeit a short but constant RR interval in the RR interval trend, histogram, and poincare plot. Furthermore, it was observed that the frequency component of the constant RR interval focused on the low frequency side on the PSD. With respect to the bigeminy signal, the RR intervals corresponding to 0.6 and 0.9 s were irregularly repeated from 0 to 17 min, and the RR intervals corresponding to 0.6 and 1.3 s were regularly repeated from 17 to 21 min. Multiple RR intervals repeated in the bigeminy signal can be identified in the histogram. The histogram of a normal ECG signal exhibited a bell-shaped RR distribution with a constant RR interval whereas the bigeminy’s histogram exhibited a multiple bell-shaped distribution with multiple RR intervals. Thus, the power of the high frequency band corresponding to 100–160 Hz increased on the PSD. The poincare plot also exhibited high distributions at 0.6, 0.9, and 1.3 s. With respect to AF patients, a slightly constant 1-s RR interval was maintained due to antiarrhythmic medication. Furthermore, constant RR intervals are observed in the RR interval trend, histogram, and poincare plot. Nevertheless, the AF-specific oscillation signal exhibited intermittent RR distributions in the histogram and considerably high power in the broad high frequency region on the PSD.

### 4.3. Operation Experiment between VSS and VB

The interworking experiments of VSS and VB were performed via the following scenario. The VB collected vital sign information from each patient in real-time. The acquired vital sign information was displayed on the LCD monitor of the VB and central monitor and was stored on the server in real-time. Furthermore, after analyzing each biometric information, an alarm was displayed on the central monitor if the information was not normal. Figure 16 shows the collected vital sign information and its upper and lower limit for alarm on the central monitor. The upper and lower limits of the alarm were set in the System Setup-Alarm menu described in Figure 5. The ECG analysis report was provided at the request of the physician.

First, we verified whether the vital sign viewer program was working properly. The login was performed by running the GUI that can connect to the VSS system, as shown in Figure 17a. By choosing the specific patient window, the patient currently monitored by the VB was connected. It was confirmed that the measured bio-signals were displayed normally in the Patient Window of the connected patient. In the vital sign viewer GUI, the patient for viewing the vital sign trend graph was selected. It was verified whether the information to be verified among the bio-signal information of the chosen patient is displayed on the vital sign viewer. As shown in Figure 17b, when entering the vital sign trend menu, the patient to be searched is selected through the patient selection popup. Additionally, the Trend sub-menu on the upper tab was selected, and the period and bio-signals to be searched were set on the displayed screen. To verify the DBMS access using MySQL, it was verified that the MySQL server and MySQL database script were installed after running the VSS system. Additionally, as shown in Figure 17c, the server start message (Server Begin Succ) is confirmed after running the VSS. The VSS executes a query to obtain the VB list when it is executed. By running the MySQL workbench program, the table and trigger configuration of the database was verified.

It was verified whether real-time bio-signal data is output from the vital sign. As shown in Figure 18a, a patient for connecting to monitor real-time data is connected and the real-time bio-signal data of the connected patient was displayed. The measured bio-signal data was also stored in the directory set up in the VSS’s configuration (ServerConfig.ini file in the figure). As shown in Figure 18b, the DP screen displays the detailed patient window for the patient’s bio-signal data. In order to verify the arrhythmia detection alarm function, the ECG signal generation simulator (Contec MS-400 [41]) was connected to the VB. Subsequently, an arrhythmia signal was generated in the simulator. If an unusual (arrhythmia suspicious) waveform was detected, the arrhythmia detection and alarm were verified in the vital sign viewer of the VSS system. Figure 18c shows an arrhythmia detection scene using the ECG simulator. The dummy VB transmits virtual measurement data upon successful login to the VSS. An arrhythmia waveform was generated through the ECG simulator linked to the VB. The data loaded by the dummy VB contained an arrhythmia waveform, and it was verified as to whether the alarm was generated for the arrhythmia waveform. The alarm was displayed in the graph area where the ECG Lead data was displayed. The interlocking experiment of the VSS and VB verified that vital sign information measured from each patient was transmitted in real-time through the VB, and the transmitted vital sign information was stored and analyzed in real-time on the server.

### 4.4. Field Test

The proposed u-Vital system is under the approval of 2nd-grade medical device by Korea Food and Drug Administration. The system is applied to the artificial kidney room of Yeongcheon hospital of Yeungnam University in South Korea and electrocardiogram and pulse of 16 patients are measured and stored in the database, as shown in Figure 19. The vital sign viewer displays a usage list of the VPs for patient information and search duration.

## 5. Discussion

PMDs using Bluetooth or including a long-term ECG analysis function in a VSS are not developed yet. The effectiveness of PMD can be maximized by replacing the existing wired-based PMD with WiFi, Bluetooth-based wireless VB, and VSS systems. The developed u-Vital system overcomes the disadvantage of not being able to store the full-wave bio-signals, and this corresponds to the inconvenience of the existing PMS. Additionally, the developed system can be used as a dedicated bio-signal measuring device by simply changing the GUI by a user, and it can be utilized as a clinical test equipment because it is easy to measure vital signs of subjects during clinical trials.

Table 3 shows the functional comparison between the proposed u-Vital system and existing PMS platforms (B40 of GE [42], IntelliVue MX40 of Philips [43], Life Scope TR of Nihon Kohden [44], and Infinity M300 of Draeger [45]). The advantage of the proposed u-Vital system is that it can wirelessly store the full-waveforms of various bio-signals compared to the conventional PMS platforms, and can analyze the ECG data in real time. It was also experimentally confirmed that at least 48 VBs can be connected to one VSS. This means that at least 48 patients can be connected to one VSS and monitored at the same time. Each VB is connected to the VSS’s wireless LAN card via WiFi, and the patient connected to each VB is simultaneously monitored via the central monitor of the VSS. Its biometric modules are wirelessly connected to the VB, taking into account the patient’s mobility.

The proposed u-Vital system can solve the disadvantages of the existing PMD, i.e., inconveniences of mobility and portability, via WiFi, and it also enables centralized monitoring of patients, thereby solving labor-intensive problems of nursing personnel. Furthermore, it can be utilized as a dedicated bio-signal meter for point of care because the GUI of the VB can be changed by a user. It can also replace the existing Holter-Recorder because it stores ECG data for 24 h via incorporating a long-term ECG analysis function in the VSS and analyzing all patient data for broader diagnostics. Additionally, it is compatible with the EMR system used in the hospital such that the patient bio-signals can be stored in the hospital EMR system without any additional equipment. The VB includes built-in WiFi and Bluetooth and can be used as a gateway for future remote medical systems.

The proposed VB and VSS system is classified as 2nd grade in medical device item permission, and this enables rapid commercialization. It can be sold in the existing domestic and foreign low-performance PMD markets and also can create a new market of short-term PMD used in various specialized disease centers such as artificial kidney rooms and endoscopy centers.

In the future, we are considering blockchain technology to build a security system for medical data in the developed system. When blockchain technology is applied to patient monitoring, measured patient bio-signal data can be safely shared between hospitals [46]. History sharing of patient bio-signals can reduce medical costs and diagnostic errors by avoiding unnecessary duplicate testing and prescriptions. In IoMT security, blockchain can prevent the manipulation of personal health records by providing patient data to hospitals and insurance companies as encrypted hashes [47]. In addition, medical institutions can collect medical records from multiple hospitals and conduct precise care that is appropriate for their patients. An important key to the application of blockchain technology in the field of IoMT is the standardization of medical data to be exchanged [48]. Currently, many medical data in South Korea are for insurance benefits, so their storage and utilization are less valuable. Therefore, we plan to develop applications that standardize patient biometric data to make patient records easy to use in any medical institution.

## 6. Conclusions

The proposed portable u-Vital system developed in the study consists of plurality of VBs for measuring bio-signals of a patient and a VSS to store, analyze, and manage the measured bio-signals. The VSS uses MySQL to store bio-signals measured from patients, and thus past and currently measured data can be efficiently managed. The portable u-Vital system can assist caregivers and medical staff in hospitals and emergency rooms where the vital signs of patients require continuous observation. The IoMT-based remote PMS can constitute an important function of future healthcare systems. Additionally, the importance of algorithms for effective alarms on measured bio-signals will also increase. In the future, we will use bio-certification and blockchain technology for the security of measured patient bio-signals. In addition, in order to determine which of the numerous medical data are effective for treatment, the priority of the medical data will be considered for the security system of the medical data.

## Figures and Tables

**Figure 1 sensors-20-01089-f001:**
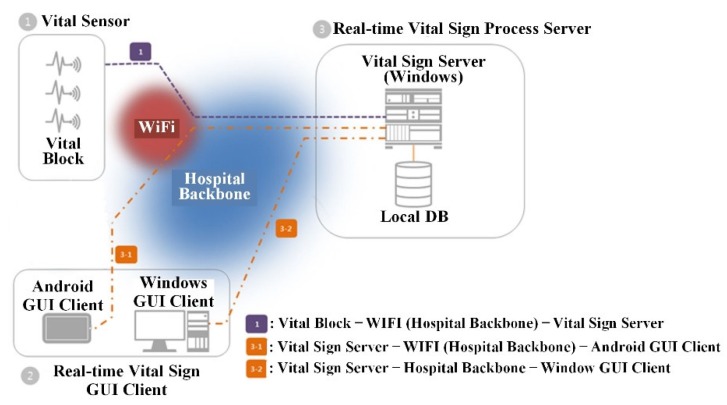
Connection configuration of Vital Block (VB) and Vital Sign Server (VSS).

**Figure 2 sensors-20-01089-f002:**
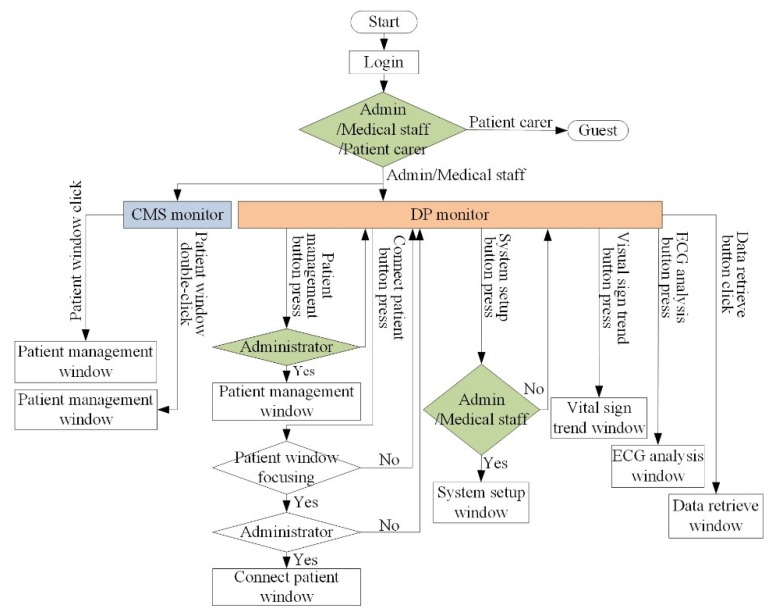
Flow chart for dual-screen of the central monitoring system (CMS) program in VSS.

**Figure 3 sensors-20-01089-f003:**
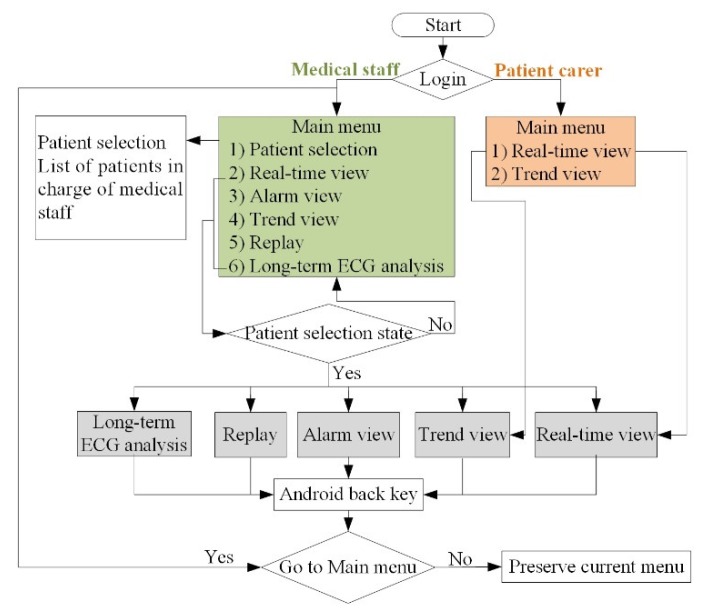
Flow chart of viewer program in Android smart device.

**Figure 4 sensors-20-01089-f004:**
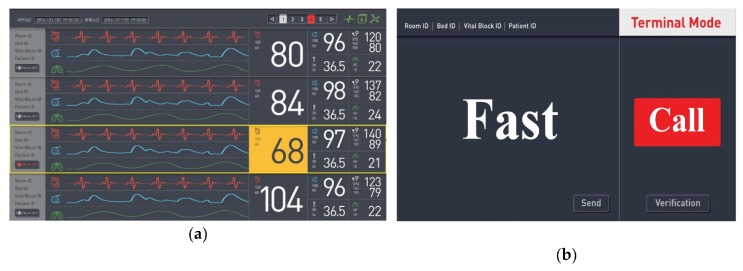
(**a**) Patient monitor (PM) mode and (**b**) patient data terminal (PDT) mode supported in the patient window.

**Figure 5 sensors-20-01089-f005:**
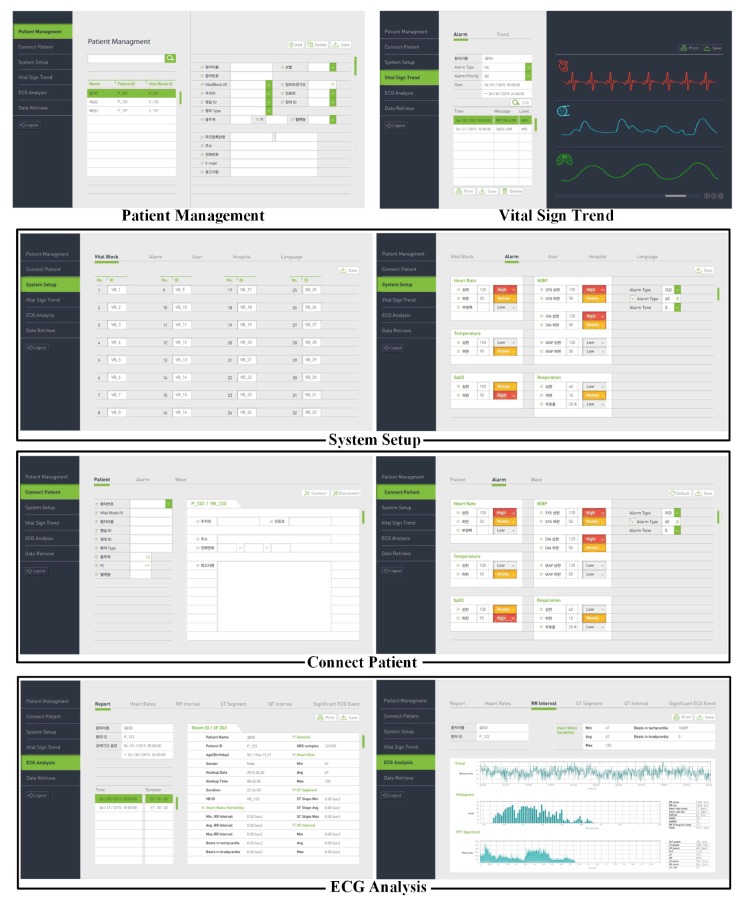
Several menus and sub-menus of the data processing (DP) program.

**Figure 6 sensors-20-01089-f006:**
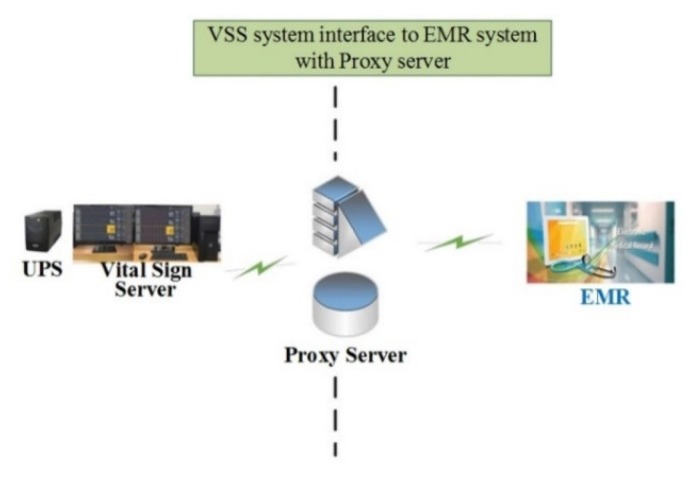
VSS system interface to electronic medical record (EMR) system with proxy server.

**Figure 7 sensors-20-01089-f007:**
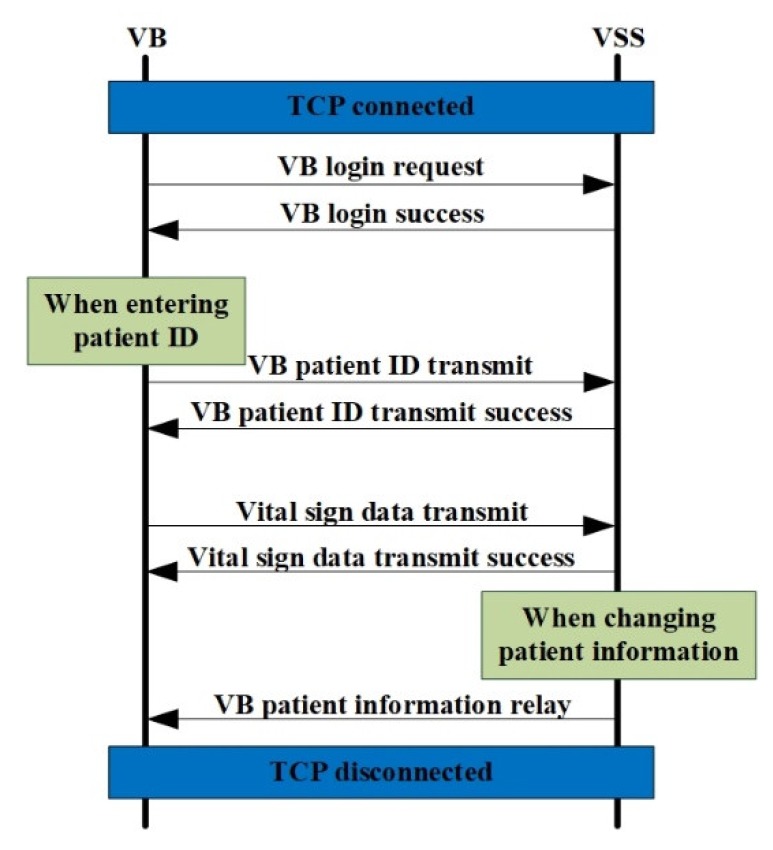
Flow chart based on usage scenario between a VB and VSS.

**Figure 8 sensors-20-01089-f008:**
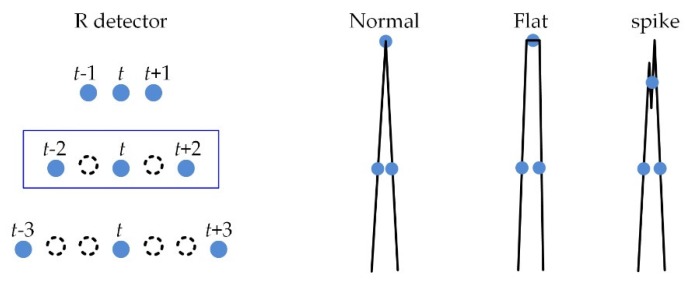
R-point detector using triangulation for various R peaks.

**Figure 9 sensors-20-01089-f009:**
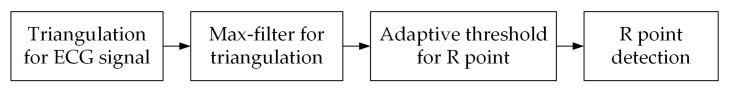
Block diagram for proposed R-point detection method.

**Figure 10 sensors-20-01089-f010:**
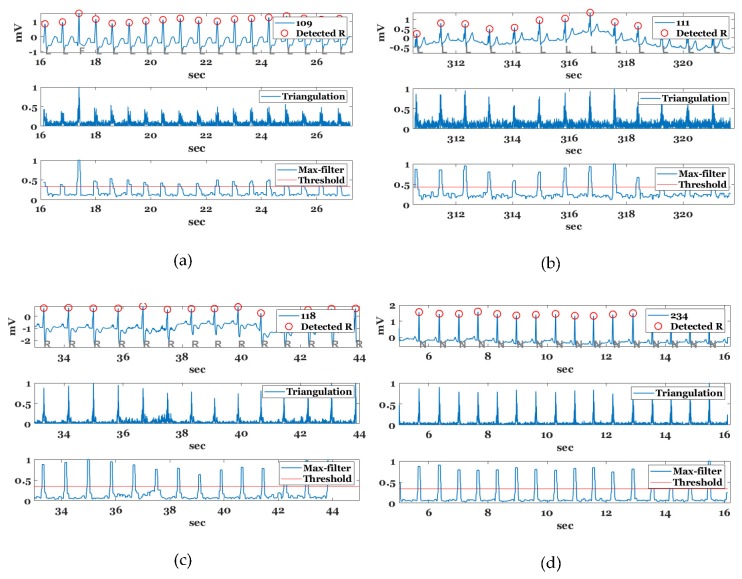
R-point detection result for Record (**a**) 109, (**b**) 111, (**c**) 118, and (**d**) 234 of MIT-BIH arrhythmia database.

**Figure 11 sensors-20-01089-f011:**
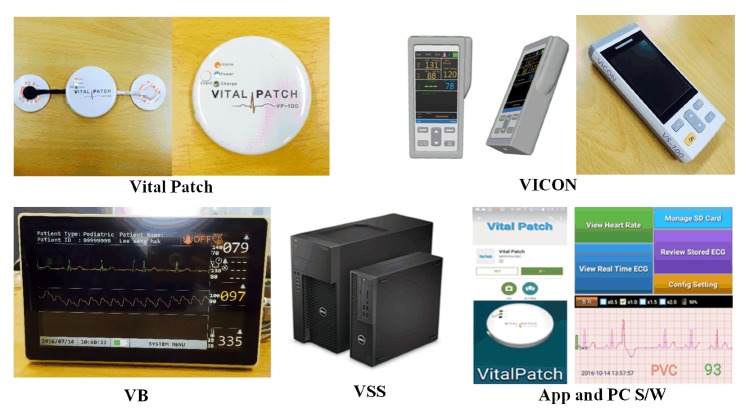
Developed vital sign measurement devices (Vital Patch (VP), VICON), VB, VSS, and App, and PC S/W).

**Figure 12 sensors-20-01089-f012:**
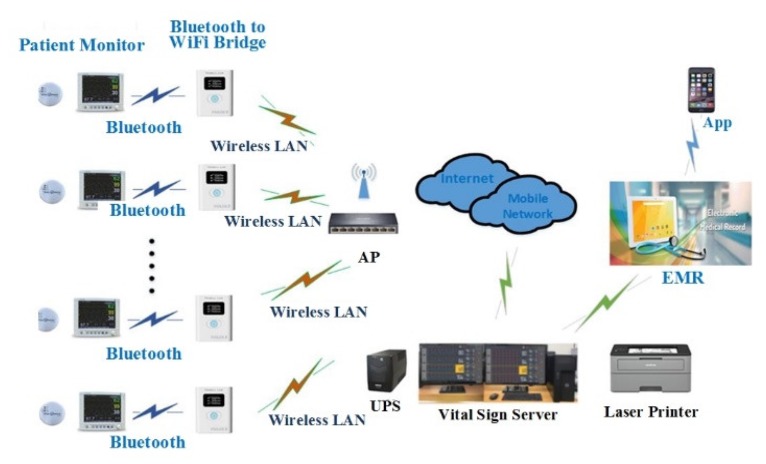
Device and system connectivity in the proposed portable u-Vital system.

**Figure 13 sensors-20-01089-f013:**
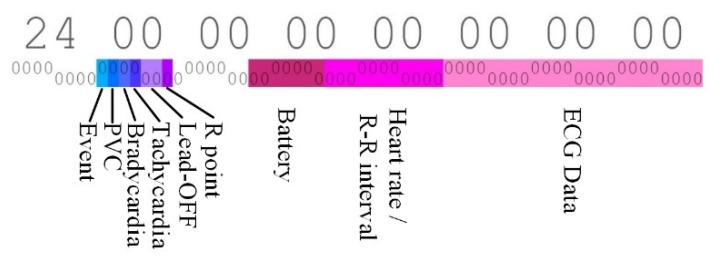
An example of data format used in the proposed u-Vital system.

**Figure 14 sensors-20-01089-f014:**
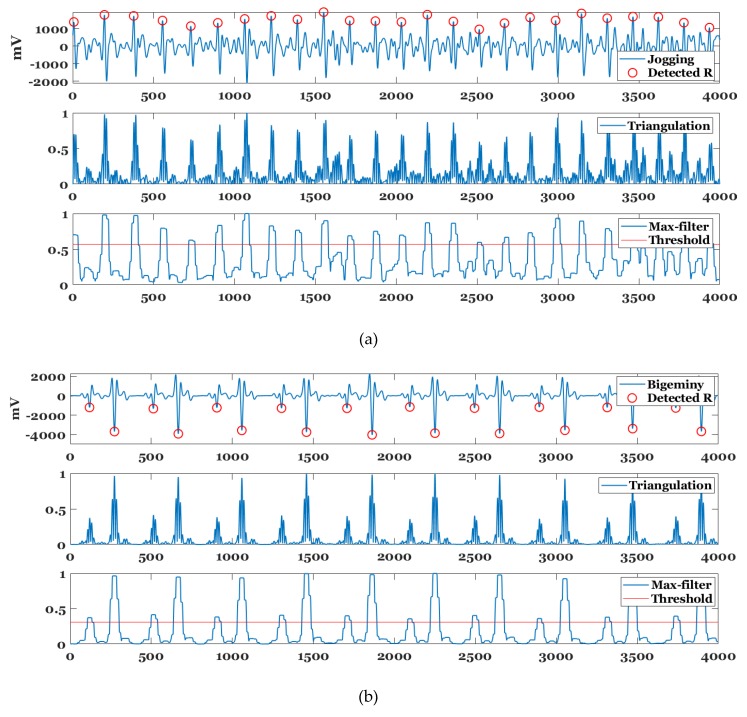
R-point detection results for (**a**) jogging, (**b**) ventricular bigeminy, and (**c**) atrial fibrillation (AF) electrocardiogram (ECG) signals.

**Figure 15 sensors-20-01089-f015:**
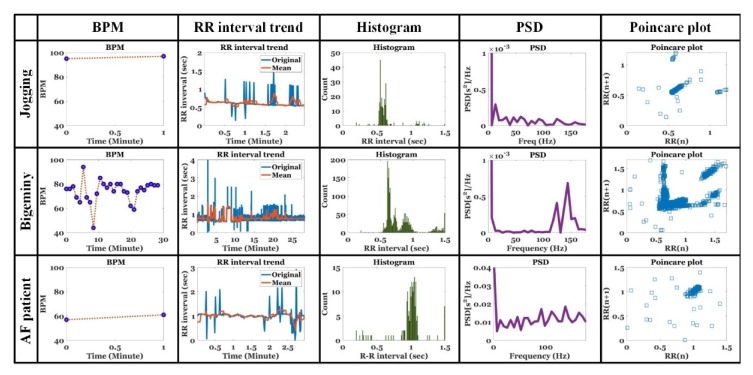
Heart rate variability (HRV) analysis for jogging, ventricular bigeminy, and AF ECG signals.

**Figure 16 sensors-20-01089-f016:**
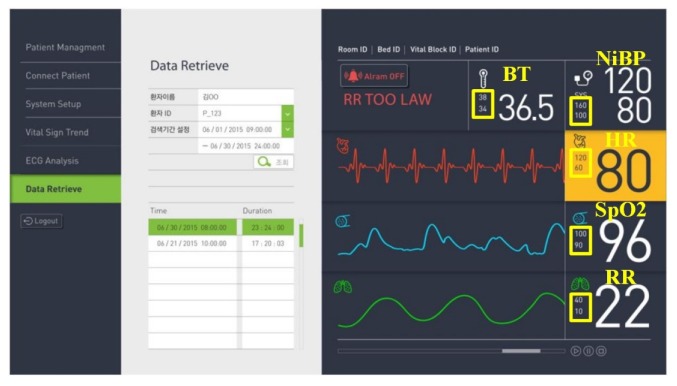
Collected vital sign information and its upper and lower limit (yellow box) for alarm on central monitor.

**Figure 17 sensors-20-01089-f017:**
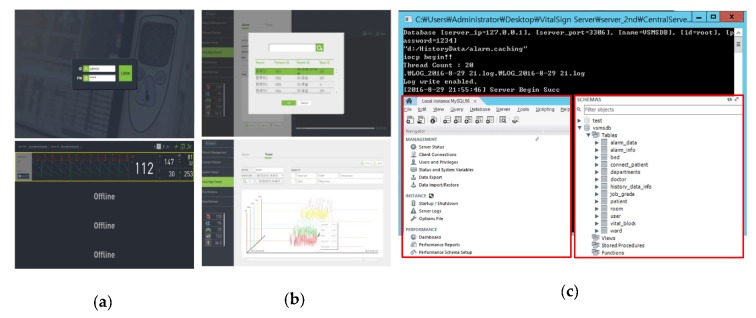
Operation check of (**a**) vital sign viewer and (**b**) vital sign trend; (**c**) configuration check of the database.

**Figure 18 sensors-20-01089-f018:**
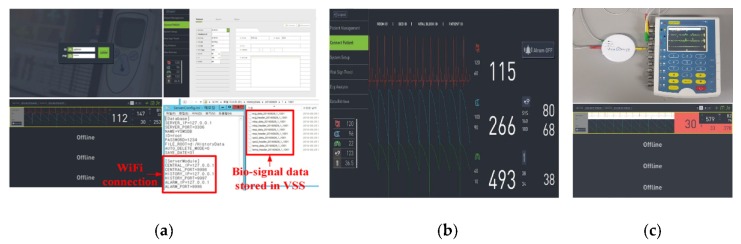
Operational verification for (**a**) checking measured real-time data, (**b**) detail monitoring screen, and (**c**) arrhythmia detection and alarm.

**Figure 19 sensors-20-01089-f019:**
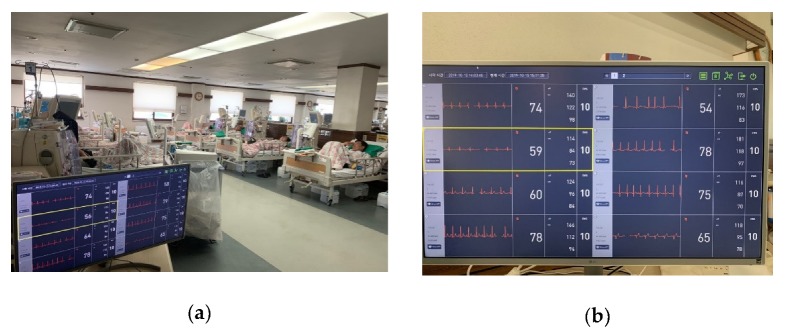
Proposed u-Vital system applied for a real hospital; (**a**) artificial kidney room and (**b**) central monitoring in VSS.

**Table 1 sensors-20-01089-t001:** R-point detection results of the proposed method using the MIT-BIH arrhythmia database.

Tape	Total	FN	FP	Se [%]	+P [%]	DER [%]	Tape	Total	FN	FP	Se [%]	+P [%]	DER [%]
100	2273	0	0	100	100	0	201	1963	9	2	99.54	99.90	0.56
101	1865	3	5	99.84	99.73	0.43	202	2136	4	3	99.81	99.86	0.33
102	2187	0	0	100	100	0	203	2980	25	16	99.16	99.46	1.38
103	2084	0	0	100	100	0	205	2656	5	2	99.81	99.92	0.26
104	2229	6	16	99.73	99.29	0.99	207	1862	6	9	99.68	99.52	0.81
105	2572	13	21	99.49	99.19	1.32	208	2955	14	6	99.53	99.80	0.68
106	2027	2	4	99.90	99.80	0.30	209	3004	2	3	99.93	99.90	0.17
107	2137	3	0	99.86	100	0.14	210	2650	22	8	99.17	99.70	1.13
108	1774	9	25	99.49	98.60	1.92	212	2748	2	3	99.93	99.89	0.18
109	2532	3	0	99.88	100	0.12	213	3251	2	7	99.94	99.79	0.28
111	2124	2	2	99.91	99.91	0.19	214	2265	1	2	99.96	99.91	0.13
112	2539	0	1	100	99.96	0.04	215	3363	2	2	99.94	99.94	0.12
113	1795	1	0	99.94	100	0.06	217	2209	7	3	99.68	99.86	0.45
114	1879	2	3	99.89	99.84	0.27	219	2154	1	3	99.95	99.86	0.19
115	1953	0	0	100	100	0	220	2048	1	0	99.95	100	0.05
116	2412	12	6	99.50	99.75	0.75	221	2427	4	1	99.84	99.96	0.21
117	1535	0	0	100	100	0	222	2483	6	1	99.76	99.96	0.28
118	2278	1	1	99.96	99.96	0.09	223	2605	4	2	99.85	99.92	0.23
119	1987	0	0	100	100	0	228	2053	9	12	99.56	99.42	1.02
121	1863	5	1	99.73	99.95	0.32	230	2256	2	2	99.91	99.91	0.18
122	2476	0	0	100	100	0	231	1571	2	1	99.87	99.94	0.19
123	1518	2	1	99.87	99.93	0.20	232	1780	2	3	99.89	99.83	0.28
124	1619	1	0	99.94	100	0.06	233	3079	4	2	99.87	99.94	0.19
200	2601	3	11	99.88	99.58	0.54	234	2753	2	1	99.93	99.96	0.11
							**Total**	**109510**	**206**	**191**	**99.81**	**99.83**	**0.36**

**Table 2 sensors-20-01089-t002:** Performance index comparison of the proposed method and the existing methods.

Method	TB	FN	FP	Se [%]	+P [%]	DER [%]
Pan et al. [9]	116,137	277	507	99.76	99.56	0.68
Benitez et al. [10]	-	203	187	99.81	99.83	0.36
Cristov [11]	110,050	240	239	99.78	99.78	0.44
Zhang et al. [12]	109,510	213	201	99.81	99.80	0.39
Castells-Rufas et al. [13]	109,494	614	353	99.43	99.67	0.88
Merah et al. [14]	109,494	178	126	99.84	99.88	0.28
Kim et al. [15]	109,494	107	97	99.90	99.91	0.19
**Proposed**	109,510	206	191	99.81	99.83	0.36

**Table 3 sensors-20-01089-t003:** Comparison of the proposed u-Vital system and existing platforms.

Functions	B40 (GE)	IntelliVue MX40 (Philips)	Life Scope TR (Nihon Kohden)	Infinity M300 (Draeger)	Proposed u-Vital System
ECG	O	O	O	O	O
SpO2	O	O	O	O	O
NiBP	O	O	O	X	O
BT	O	X	O	X	O
BIS	O	X	O	X	X
Connection with modules	wire	wire	wire	wire	wireless
Wireless network (Gateway)	O	X	X	O	O
Num. of connected patients	-	1	16	1	48
Arrhythmia alarm	O	X	O	O	O
HRV	X	X	X	X	O
Storage of full-waveforms	X	X	X	X	O

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
