# Peer review of "Vital Block and Vital Sign Server for ECG and Vital Sign Monitoring in a Portable u-Vital System"

_sensors, 2020, doi:10.3390/s20041089_

Round 1

Reviewer 1 Report

Internet of Medical Things (IoMT) is a very recent research area that refers to the connected system of medical devices and applications that collect data that is then provided to healthcare IT systems through online computer networks.

The contribution of the paper is twofold: 1) description, design and implementation of a novel IoMT platform (u-Vital) and 2) implementation of an R-point detection algorithm for real-time processing and long-term ECG analysis.

The novelty of the reserarch is not high as there are already several platforms capable of monitoring vital parameters and transmitting them to a server in order to save data and manage the information for further offline analysis (ex. s OpenEHR, eVisit, ...). An innovative aspect in this sense could be to integrate both wearable and environmental sensors within the platform, also to expand the number of end-users (the acceptability of the solution could be increased, not all end-users appreciate wearable sensing solutions). 

In the introduction the authors compare existing QRS or R-point detection methods already present in the literature, but the same work has not been done for ioMT solutions/platforms and in my opinion this is necessary to highlight the innovative aspect of the proposed system compared to the existing ones.

Moreover, I believe it is necessary in the conclusions section to better highlight the future developments of this research activity.

Some questions for the authors: 

the platform provide mechanisms for data sharing with third party applications? 

Have you considered in your system same aspect related to IoMT security vulnerabilities?

Some minor issues: a) acronyms are used in the paper without having been first introduced, b) improve the quality of Figure 5. 

Author Response

Please note that English has been corrected for the entire paper. Thank you.

Internet of Medical Things (IoMT) is a very recent research area that refers to the connected system of medical devices and applications that collect data that is then provided to healthcare IT systems through online computer networks.

The contribution of the paper is twofold: 1) description, design and implementation of a novel IoMT platform (u-Vital) and 2) implementation of an R-point detection algorithm for real-time processing and long-term ECG analysis.

The novelty of the reserarch is not high as there are already several platforms capable of monitoring vital parameters and transmitting them to a server in order to save data and manage the information for further offline analysis (ex. s OpenEHR, eVisit, ...). An innovative aspect in this sense could be to integrate both wearable and environmental sensors within the platform, also to expand the number of end-users (the acceptability of the solution could be increased, not all end-users appreciate wearable sensing solutions).

In the introduction the authors compare existing QRS or R-point detection methods already present in the literature, but the same work has not been done for ioMT solutions/platforms and in my opinion this is necessary to highlight the innovative aspect of the proposed system compared to the existing ones.

Ans) The advantages of the proposed u-Vital system are added to the Introduction and Discussion as follows.

--------------------------------------------------------------------------------------------------------------

The advantage of the proposed u-Vital system is that it can wirelessly store the full wave of various bio-signals compared to the conventional PMS, and can analyze the ECG data in real time. It was also experimentally confirmed that at least 48 VBs can be connected to one VSS. This means that at least 48 patients can be connected to one VSS and monitored at the same time.

Moreover, I believe it is necessary in the conclusions section to better highlight the future developments of this research activity.

Ans) The following medical data security plan has been added to the conclusion.

In the future, we will use bio-certification and blockchain technology for the security of measured patient bio-signals. In addition, in order to determine which of the numerous medical data are effective for treatment, the priority of the medical data will be considered for the security system of the medical data.

Some questions for the authors:

the platform provide mechanisms for data sharing with third party applications?

Ans) Many patient monitoring systems currently use different data formats, making medical data less compatible. Therefore, standardization of medical data is urgent as mentioned section 2. The developed system also uses the data format we defined. The following Figure 13 and content have been added.

-------------------------------------------------------------------------------------------------------------

Many patient monitoring systems currently use different data formats, making medical data less compatible. Therefore, standardization of medical data is urgent as mentioned section 2. The developed system also uses the data format we defined. Figure 13 shows an example of the data format of the proposed u-Vital system. 1 bit is assigned to Event, PVC, Bradycardia, Tachycardia, and R point flags, and 2 bits are assigned to Lead-off. In addition, 7 bits for battery information, 11 bits for heartrate and RR interval, and 24 bits for ECG data are allocated.

Figure 13. An example of data format used in the proposed u-Vital system.

Have you considered in your system same aspect related to IoMT security vulnerabilities?

Ans) We are considering IoMT security as a future work. The following has been added to Discussion.

--------------------------------------------------------------------------------------------------------------

In the future, we are considering blockchain technology to build a security system for medical data in the developed system. When blockchain technology is applied to patient monitoring, measured patient bio-signal data can be safely shared between hospitals [42]. History sharing of patient bio-signals can reduce medical costs and diagnostic errors by avoiding unnecessary duplicate testing and prescriptions. In IoMT security, blockchain can prevent the manipulation of personal health records by providing patient data to hospitals and insurance companies as encrypted hashes [43]. In addition, medical institutions can collect medical records from multiple hospitals and conduct precise care that is appropriate for their patients. An important key to the application of blockchain technology in the field of IoMT is the standardization of medical data to be exchanged [44]. Currently, many medical data in South Korea are for insurance benefits, so their storage and utilization are less valuable. Therefore, we plan to develop applications that standardize patient biometric data to make patient records easy to use in any medical institution.

[42] Seliem, M.; Elgazzar K. BIoMT: blockchain for the internet of medical things. IEEE International Black Sea Conference on Communications and Networking, Sochi, Russia, 3-6 June 2019.

[43] Vazirani, A.A.; Donoghue, O.; Brindley, D.; Meinert, E. Blockchain vehicles for efficient medical record management. Digital Medicine 2020, 3, 1-5.

[44] Dimitrov, D. Blockchain applications for healthcare data management. Healthc Inform Res. 2019, 25(1), 51–56.

And the following is added to the conclusion:

--------------------------------------------------------------------------------------------------------------

In the future, we will use bio-certification and blockchain technology for the security of measured patient bio-signals. In addition, in order to determine which of the numerous medical data are effective for treatment, the priority of the medical data will be considered for the security system of the medical data.

Some minor issues: a) acronyms are used in the paper without having been first introduced, b) improve the quality of Figure 5.

Ans) It was modified as follows and we improved Figure 5.

ODM and OEM -> as original development manufacturing and original equipment manufacturing for PMDs of first-rate companies

Reviewer 2 Report

See attached file

Author Response

Please note that English has been corrected for the entire paper. Thank you.

The paper proposes a patient-monitoring system consisting of smart devices and IT technology to collect and analyze several vital signs (such as ECG, blood pressure, oxygen saturation and body temperature) in the context of IoMT applications, that can be stored and managed remotely. The system aims to overcome the burden of wired devices through wireless connections and real-time transmission of bio-signals to a centralized server capable of simultaneously monitoring up to 32 patients. In general, the paper is interesting as well as the advantages of using the system in hospital setting, in order to improve efficiency in patient care and face the next challenges for the healthcare systems. However, I have some doubts about the paper, which is unclear in some points and, in particular, does not appear to have a well-defined trait. On the one hand, the authors describe the complex system from a technical point of view (architecture, devices, protocols, communication), sometimes going into too much detail (as for the description of the GUI). On the other hand, they also propose a new method to detect R-points in the ECG signal, including the performance on ECG recordings of a reference dataset, the comparison with other similar works and the results on three experimental cases. The result is a paper that, at the same time, is not completely technical nor completely methodological, remaining at the moment, and in my opinion, incomplete in both contexts. In addition to these personal and general observations, some comments that should however contribute to improve the quality and readability of the paper.

MAJOR:

L44: Why is SQLite database mentioned? Does this sentence refer to the previous work cited ([8]) work example) to avoid misunderstandings.

Ans) We don't think it's necessary in context. So it (SQLite) was deleted.

L77-L82: add further references to other works for the mentioned biometric parameters, connection protocols and OS.

Ans) We have included the following references (biometric parameters [16, 17], connection protocols [19, 20] and OS [21, 22]) in our paper:

[16] Dias, D.; Cunha, J.P.S. Wearable health devices-vital sign monitoring, systems and technologies. Sensors 2018, 18(8), 1-28.

[17] Alhayajneh, A.; Baccarini, A.N.; Weiss, G.M.; Hayajneh, T.; Farajidavar, A. Biometric authentication and verification for medical cyber physical systems. Electronics 2018, 7(12), 1-17.

[19] Shobha, G.; Chittal, R.R.; Kumar, K. Medical applications of wireless networks. In Proceedings of the 2nd International Conference on Systems and Networks Communications, Cap Esterel, France, 25-31 Aug. 2007.

[20] Lai, D.T.H.; Begg, R.K.; Palaniswami, M. Sensor Networks in healthcare: a new paradigm for improving future global health. In Healthcare Sensor Networks: Challenges Toward Practical Implementation, 1st ed.; Lai, D.T.H., Begg, R.K., Palaniswami, M.; CRC Press: New York, USA, 2011; pp. 9–11.

[21] Schobel, J.; Schickler, M.; Pryss, R.; Nienhaus, H. Using vital sensors in mobile healthcare business applications: challenges, examples, lessons learned. In Proceedings of the 9th International Conference on Web Information Systems and Technologies, Aachen, Germany, 8-10 May 2013.

[22] Cornet, V.P.; Holden, R.J. Systematic review of smartphone-based passive sensing for health and wellbeing. Journal of Biomedical Informatics 2018, 77, 120-132.

L89-L100: add further references to other works for the points mentioned.

Ans) The following references were added.

[27] Khalifa, M.; Alswailem, O. Hospital information systems (HIS) acceptance and satisfaction: a case study of a tertiary care hospital. Procedia Computer Science 2015, 63, 198-204.

[28] Kuo, K.M.; Liu, C.F.; Talley, P.C.; Pan, S.Y. Strategic improvement for quality and satisfaction of hospital information systems. Journal of Healthcare Engineering 2018, 2018, 1-14.

[29] OEM/ODM. Available online: https://micromaxhealth.com/products/oem-odm/ (accessed on 30 Jan. 2020).

[31] Feldman, K.; Johnson, R.A.; Chawla, N.V. The State of data in healthcare: path towards standardization. Journal of Healthcare Informatics Research 2018, 2(3), 248-271.

[32] Grannis, S.J.; Xu, H.; Vest, J.R.; Kasthurirathne, S.; Bo, N.; Moscovitch, B.; Torkzadeh, R.; Rising, J. Evaluating the effect of data standardization and validation on patient matching accuracy. Journal of the American Medical Informatics Association 2019, 26, 447-456.

[33] Smith, M.; Hoka, S.; Farag, E. BiSpectral Index (BIS) monitoring may detect critical hypotension before automated non-invasive blood pressure (NIBP) measurement during general anaesthesia; a case report. F1000Research 2014, 3(5), 1-5.

L101: a reference to the classification of PMS based on performance is missing

Ans) We added the following references.

[34] Patient monitor. Available online: http://www.bistosamerica.com/modules/catalogue_eng2/cg_view.html?cc=12&PHPSESSID=93cb2bdaeed208f40b3bbd8ddef32a8d&p=1&no=21&PHPSESSID=93cb2bdaeed208f40b3bbd8ddef32a8d/ (accessed on 30 Jan. 2020).

[35] Respiratory rate patient monitor/ETCO2/SPO2/intensive care. Available online: https://www.medicalexpo.com/prod/meditech-equipment/product-80924-760194.html/ (accessed on 30 Jan. 2020).

[36] Vital signs monitor ECG, SPO2, NIBP, RESP, TEMP and CO2 capnography. Available online: https://www.quirumed.com/en/vital-signs-monitor-ecg-spo2-nibp-resp-temp-and-co2-capnography.html/ (accessed on 30 Jan. 2020).

L125: The configuration of the u-Vital System is unclear: does it use a single VB (which is a PMD)? But, according to the classification in L101-109, different PMDs measure different bio-signals, therefore the u-Vital System should be composed of several PMDs (in section 4, both Vital Patch and VICON are mentioned). The authors should clarify this point.

Ans) Yes, right. At least 48 VBs can be connected to one Vital Sign Server and one VB is allocated per patient. These were added to the L133.

L131: What does mean? Are other bio-signals monitored by your system?

Ans) It means vital signals measured from VB. We corrected it at L138.

L164: Who is a protector? What is its role in the monitoring system? More details should be included.

Ans) It means patient carer. It was corrected in L171, Figure 2 and 3. L171 has been revised as follows: If the logged-in user is a patient carer, only two menus (Real-time view and Trend view) are available for providing basic bio-signal information measured for the patient.

L187-221: The description of the GUIs is too detailed, like a sort of user manual: in my opinion, this is important for a user or operator but not for readers. It should be better to explain the main features provided by the GUIs, keeping a more general description.

Ans) We have shortened the original paragraph and added a description of some key features at L194~L220.

L226: The database used to store data is misleading: MySQL is indicated in L148, here MariaDB is indicated. How many DBs are used by the system? What is the function of Oracle DB?

Ans) MySQL database is applied to this system. After that it is upgrading to MariaDB. In this paper, it is referred to as MySQL, the original database.

L241: Which “users” will use the system? Is it intended only for clinical personnel or also for domestic use?

Ans) This system is designed for use in hospitals. So the user is a medical staff. At L239, the user was changed to a medical staff.

L242: Something is wrong with the sentence: the VSS should not respond “successful login request”? to “patient ID request”

Ans) VSS always responds to VB's login request and patient ID transmission. The sentence has changed to: the VSS accepts a successful login request to the VB at L240.

L244: What does “predetermined cycle” mean? How often does the transmission take place? Further details on the timing of request/response messages should be added.

Ans) The following has been added at L242.

The bio-data transmission/reception time between the VP and VSS = 0.5 s

L264-272: The sequence of steps for the detection of R-points should be better explained by including some missing information (for example sliding window size, description of the max-filter).

Ans) The following has been added to L264 ~ L269.

The max filter defines a candidate interval of the R-point detection for the enhanced R wave by the triangulation processing. The size of the max filter used for MIT-BIH arrhythmia database (360 samples/sec) [40] is 1×25 sample interval. The size of the sliding window and step size were set to 4000 and 500 samples respectively based on MIT-BIH arrhythmia database.

L297: In table 1, the results with the proposed method seem to indicate a variable behaviour in the detection of R-points: it passes from 0 FP/FN (as in record 102) to 41 FP/FN (as in record 203). Do the authors have an explanation or hypothesis for this behaviour? Does it depend on specific characteristics of the analysed ECG recordings?

Ans) The following has been added to L294 ~ L296.

Record 203 has a high false alarm rate because it has many arrhythmic rhythms such as atrial flutter, atrial fibrillation, ventricular trigemini, and ventricular tachycardia compared to other records.

L299: What is TB in Table 2? It was not considered in Table 1. Do the results of other studies refer to the same records analysed? Are the results really comparable because they are obtained under the same analysis conditions?

Ans) TB means total beats of all the Records of the MIT-BIH arrhythmia database. It is added at L298-L299.

L311: Body temperature (BT) signal is missing

Ans) Thank you. It is added at L316.

L319: Information on the time between VICON and VSS is missing.

Ans) Thank you. It is added at L324.

L324: The experimental conditions should be indicated more precisely: have the ECG signals been acquired via Virtual Patch? Have ECG signals been analysed, in real-time, by the proposed R-point detection method?

Ans) The actual ECG data was collected by the vital patch and analyzed in real time by the proposed method. This has been added to the L338 as follows:

Actual ECG data was measured from the VP with a 250 Hz sampling rate and analyzed throughout HRV by the proposed R-point detection method.

L361-366: Several bio-signals can be collected by the system (as happens in IoMT applications), but the general description and analysis seem to focus only on the ECG. Why? What about other bio-signals? What about the alarms that can be set in the system?

Ans) Since ECG signals are more difficult to analyze than other bio-signals, we have explained more about them. We have added descriptions of the different bio-signals and their alarms at L382 ~ 384 and Figure 16 as the following.

Figure 16 shows the collected vital sign information and its upper and lower limit for alarm on the central monitor. The upper and lower limits of the alarm were set in the System Setup-Alarm menu described in Figure 5.

Figure 16. Collected vital sign information and its upper and lower limit (yellow box) for alarm on central monitor.

L367-416: As in point 8, the description is too detailed and more addressed to operators rather than to readers. Furthermore, MySQL DB (and not MariaDB) has been re-indicated causing possible confusion about the type of database used.

Ans) Unnecessarily much information has been reduced as shown at L388 ~ L402 and MariaDB has been removed.

Discussion and Conclusions could be extended.

Ans) The following contents were added in Discussion and Conclusions.

(Discussion) In the future, we are considering blockchain technology to build a security system for medical data in the developed system. When blockchain technology is applied to patient monitoring, measured patient bio-signal data can be safely shared between hospitals [42]. History sharing of patient bio-signals can reduce medical costs and diagnostic errors by avoiding unnecessary duplicate testing and prescriptions. In IoMT security, blockchain can prevent the manipulation of personal health records by providing patient data to hospitals and insurance companies as encrypted hashes [43]. In addition, medical institutions can collect medical records from multiple hospitals and conduct precise care that is appropriate for their patients. An important key to the application of blockchain technology in the field of IoMT is the standardization of medical data to be exchanged [44]. Currently, many medical data in South Korea are for insurance benefits, so their storage and utilization are less valuable. Therefore, we plan to develop applications that standardize patient biometric data to make patient records easy to use in any medical institution.

(Conclusion) In the future, we will use bio-certification and blockchain technology for the security of measured patient bio-signals. In addition, in order to determine which of the numerous medical data are effective for treatment, the priority of the medical data will be considered for the security system of the medical data.

MINOR:

L50: The full name for QRS should be entered.

Ans) It is commonly used as the Q, R, and S wave sections of ECG.

L93: Explain the meaning of ODM and OEM.

Ans) It was modified as follows.

ODM and OEM -> as original development manufacturing and original equipment manufacturing for PMDs of first-rate companies

L106: Probably, PMD (device) instead of PMS (system)

Ans) We corrected it.

Round 2

Reviewer 1 Report

I would further highlight the innovative aspects of the platform compared
to existing platforms, otherwise the corrections made are sufficient
for publication

Author Response

I would further highlight the innovative aspects of the platform compared to existing platforms, otherwise the corrections made are sufficient for publication.

Ans) The advantages of the proposed system over the existing platform are included in Discussion, with Table 3.

Table 3 shows the functional comparison between the proposed u-Vital system and existing PMS platforms (B40 of GE [42], IntelliVue MX40 of Philips [43], Life Scope TR of Nihon Kohden [44], and Infinity M300 of Draeger [45]). The advantage of the proposed u-Vital system is that it can wirelessly store the full-waveforms of various bio-signals compared to the conventional PMS platforms, and can analyze the ECG data in real time. It was also experimentally confirmed that at least 48 VBs can be connected to one VSS. This means that at least 48 patients can be connected to one VSS and monitored at the same time. And its biometric modules are wirelessly connected to the VB, taking into account the patient's mobility.

Table 3. Comparison with existing platforms.

Functions

B40

(GE)

IntelliVue MX40

(Philips)

Life Scope TR

(Nihon Kohden)

Infinity M300

(Draeger)

Proposed u-Vital system

ECG

O

O

O

O

O

SpO2

O

O

O

O

O

NiBP

O

O

O

X

O

BT

O

X

O

X

O

BIS

O

X

O

X

X

Connection with modules

wire

wire

wire

wire

wireless

Wireless network (Gateway)

O

X

X

O

O

Num. of connected patients

-

1

16

1

48

Arrhythmia alarm

O

X

O

O

O

HRV

X

X

X

X

O

Storage of full-waveforms

X

X

X

X

O

Reviewer 2 Report

See comments on file

Author Response

The authors replied to my comments and observations, reviewing the manuscript in many parts, enriching the key points with new references and explanations, shortening the too descriptive paragraphs: the new version is significantly improved in terms of quality, clarity, completeness and readability. In my opinion, the manuscript should be accepted, after still fixing the following minor points (lines refer to v2 version of the manuscript)

MINOR:

1. L74-75: It is not clear how many VBs can be connected to a single VSS (and consequently how many patients are simultaneously monitored): 48 VBs is indicated here; in L148.

Ans) The content you pointed out was included in the discussion (L440~449) as the following.

Table 3 shows the functional comparison between the proposed u-Vital system and existing PMS platforms (B40 of GE [42], IntelliVue MX40 of Philips [43], Life Scope TR of Nihon Kohden [44], and Infinity M300 of Draeger [45]). The advantage of the proposed u-Vital system is that it can wirelessly store the full-waveforms of various bio-signals compared to the conventional PMS platforms, and can analyze the ECG data in real time. It was also experimentally confirmed that at least 48 VBs can be connected to one VSS. This means that at least 48 patients can be connected to one VSS and monitored at the same time. Each VB is connected to the VSS's wireless LAN card via WiFi, and the patient connected to each VB is simultaneously monitored via the central monitor of the VSS. And its biometric modules are wirelessly connected to the VB, taking into account the patient's mobility.

2. Figure 2: According to the first decision block, only Admin/Medical Staff can access the CMS and DP monitors (as correctly indicated in L163-164); but the second decision block indicates that patient carers also have access to the System setup window . Why can patient carers access the system setup window of a system designed for hospital wards? In any case, if this is correct, it would be better to modify the flowchart by duplicating the block System setup window (for patient carers ) and moving it next to GUEST terminal block.

Ans) Thank you for your point. Patient carers do not have access to the System setup window. In the second decision block of Figure 2, Patient carers was removed.

3. L291: Considering that record 203 is just an example, the sentence could be changed to something like For example, record 203 has a high .. On the contrary, record 102 . to motivate the different algorithm behaviour.

Ans) The content has been fixed at L290~293.

Some signals, such as Record 108, 203, and 210, has a high false alarm rate because they have many arrhythmic rhythms such as atrial flutter, atrial fibrillation, ventricular trigemini, and ventricular tachycardia. On the other hand, normal signals without arrhythmia rhythms such as Record 100, 102, and 103 have very high sensitivity and positive prediction.

4. L323: typo (VIOCN instead of VICON)

Ans) Thank you. It has been fixed at L323.